# When Complexity Is Good: Do We Need Recurrent Deep Learning For Time Series Outlier Detection?

## Abstract

Outlier detection is a critical part of understanding a dataset and extracting results. Outlier detection is used in different domains for various reasons; including detecting stolen credit cards, spikes of energy usage, web attacks, or in-home activity monitoring. Within this paper, we look at when it is appropriate to apply recurrent deep learning methods for time series outlier detection versus non-recurrent methods. Recurrent deep learning methods have a larger capacity for learning complex representations in time series data. We apply these methods to various synthetic and real-world datasets, including a dataset containing information about the in-home movement of people living with dementia in a clinical study cross-referenced with their recorded unplanned hospital admissions and infection episodes. We also introduce two new outlier detection methods, that can be useful in detecting contextual outliers in time series data where complex temporal relationships and local variations in the time series are important.

## 1 Introduction

Outlier detection is the process of identifying unusual data points, or collections of data points within a dataset. Within different contexts, these outliers can take various forms, for example:

- Credit card fraud detection: Unusual spending.

- Internet traffic monitoring: Unusual surges in traffic or security monitoring.

- Dementia care: Unusual patterns of activity, unusual sleep/wake patterns at night.

This paper will focus specifically on time series outliers. These are outliers that appear in sequential data, which Blázquez-García et al. (2021) argued could take three main forms:

- Point outliers: These are single data points in time that are unusual from the rest of the dataset when considering other values in the time series or points locally. Point outliers could have global unusual values, or contextually (in our case, temporally) unusual values.

- Subsequence/Collective outliers: These are consecutive points whose behaviour as a collective is unusual, but not necessarily if considered separately.

- Outlier time series: These are instances in which an entire time series (for example, a variable in a multivariate dataset) can be considered unusual.

In this paper, we focus on point and subsequence outliers and discuss when and how recurrent deep learning techniques could be used for this type of outliers compared to non-recurrent techniques. This involves testing several algorithms on multiple datasets containing different outlier types. We also introduce two new outlier detection methods by modifying the Long Short Term Memory Recurrent Neural Network (LSTM-RNN) (Hochreiter & Schmidhuber, 1997) outlier detection technique presented in Bontemps et al. (2016); Singh (2017). We change the underlying forecasting model and instead of an LSTM-RNN, we train a transformer model (Vaswani et al., 2017) and similarly, use its prediction loss as the basis for the outlier score calculation. We also removed the

focus on collective outliers (as is in the LSTM-RNN outlier detection model), so that our model was applicable to both point and collective outliers.

We continue this paper by introducing some of the preliminary concepts that will be required to design the outlier detection settings, as well as explaining the difference between recurrent and non-recurrent based outlier detection algorithms. Next, this paper discusses the related works and describes how our study extends the existing work. We introduce the methods and models that will be used to answer the questions of this paper, followed by the results and discussion on their significance.

This paper's aim is to answer the following question:

"When detecting outliers in time series data, is it appropriate to use recurrent neural networks and how do we design models for this purpose?"

Our motivation for answering this question is to enable a better understanding of which outlier detection techniques would be most usefully applied to our healthcare monitoring "real-world" time series dataset, containing movement information collected form the homes of people living with dementia (more information on Section 2.1). We also wish to understand how useful recurrent based algorithms are when points in the dataset contain complex temporal relationships with each other. For example, we wish to find outliers that are not only locally and globally unusual in their values, but where they are also unusual given some sequence that they exist in.

## 1.1 PRELIMINARY CONCEPTS

### WHAT IS A RECURRENT NEURAL NETWORK?

Within this paper, we will refer to two different types of outlier detection methods. Those that use non-recurrent techniques and those that rely on recurrent deep learning models.

Recurrent neural network approaches are characterised by their use of sequences in the time series data to detect outliers. Classical non-recurrent algorithms assume no temporal relationship between data points and calculate outlier scores on a point-by-point basis, whilst recurrent neural network algorithms use temporal relationships when detecting outliers. In theory, this difference should make recurrent neural network models superior to non-recurrent models when detecting outliers in complicated time series data. This is not necessarily the case however, since many outliers will hold unusual enough values such that they can be detected by non-recurrent techniques. In addition, extra capacity for learning complicated relationships in the data brings larger numbers of hyperparameters to tune, which can cause difficulty in training robust models. This hyper-parameter tuning becomes an issue when applying complex techniques to new datasets for which there is no outlier ground truth to measure model performance.

### OUTLIERS IN A DATASET

As discussed in Section 1, outliers take three main forms when viewed in the context of time series data. The number of outliers in a dataset is defined as a proportion of the whole dataset and is called the contamination. To contaminate a dataset is to add outliers to that dataset. When measuring the performance of outlier detection techniques, we will use a harmonic mean of precision and recall, defined as the F1 score (see Appendix A.1).

## 1.2 RELATED WORK

There has been a long history of developing outlier detection methods and exploring different algorithms for time series outlier detection (Lai et al., 2021b; Blázquez-García et al., 2021). The existing works propose various methods to identify different types of outliers that are present in time series datasets. Lai et al. (2021b) show comparisons between different algorithms for outlier detection and evaluate their performance on both synthetic and real-world datasets. However, Lai et al. (2021b) do not directly compare recurrent and non-recurrent based outlier detection algorithms.

Elsayed et al. (2021) aimed to address a similar question in relation to time series forecasting, in which they argued that a carefully configured Gradient Boosted Regression Trees (GBRT) model

(Friedman, 2001) could compete and even sometimes out-perform many complex deep learning models for time series forecasting. Pang et al. (2021) studied the role of deep learning in outlier detection, in which they acknowledge that complex models would be useful for complex anomalies (those with complicated spatial or temporal dependencies), although it is not as widely explored as evaluating the performance of complex models for detecting point anomalies. Both of these studies, however, do not directly compare recurrent and non-recurrent based algorithms.

We aim to extend the scope of Lai et al. (2021b)'s work by specifically investigating whether recurrent neural networks are useful for outlier detection compared to non-recurrent methods. An answer to this question would guide us in our search for a reliable outlier detection method for application on our remote monitoring time series dataset, related to dementia care (described in Section 2.1). We are aware that our dataset will contain complex temporal relationships between data points and we wish for an outlier detection algorithm to be proficient at finding all of these types of outliers.

## 2 METHODS AND MODELS

In this section, we discuss different datasets used in our experiments and describe how the datasets are generated or collected. Next, we discuss which existing recurrent and non-recurrent outlier detection models we have chosen. We also introduce two new outlier detection methods that rely on transformer models (Vaswani et al., 2017).

### 2.1 DATASET

To evaluate the models, we have used the TODS python package [1] and benchmarks (Lai et al., 2021a), which contain several synthetic and real-world datasets for time series outlier detection.

The synthetic datasets were generated using sinusoidal waves with various frequencies and amplitudes. These datasets are multivariate and always consist of 5 variables. Contamination functions are used to contaminate this dataset with 5 different objectives as outlined in Lai et al. (2021b). These contamination objectives are aimed at producing point outliers or collective outliers. We test the algorithms in this paper on 15 different datasets, with different combinations of types of contamination. A more thorough explanation of the way that these datasets are generated is given in Lai et al. (2021b).

The TODS package also readily makes available two real-world datasets (see Appendix A.2.1 for more information), labelled with outliers relating to the context. Those two datasets are:

- **SWAN-SF:** SWAN-SF was collected by the Harvard Dataverse and contains space weather data (Angryk et al., 2020a). This dataset has a contamination factor of $23.8\%$.
- **GECCO:** This is a dataset collected by SPOTSeven Lab [2] for a data challenge in 2018 and contains water quality information. This dataset has a contamination factor of $1.2\%$.

These datasets were processed by Lai et al. (2021b) before being made available for outlier detection. The code that processes the data is readily available in the TODS package (see Section 6 for information on reproducibility).

Alongside these datasets, we also tested different models on our dementia care dataset (referred to in this paper as the Movement dataset). This data contains movement information, collected from over 140 people living with dementia in their real home settings for a period of over 2 years (from February 2019 to June 2021). Each feature in this dataset represents the number of times a sensor was triggered in a 3 hour period in a location of the home. To simplify this, we will think of the sensor triggers as a visit to a room containing the sensor. The dataset contains 5 of these locations, which represent the hallway, living room, bedroom, bathroom and kitchen. These 3 hour time windows were then concatenated to form points that cover 24 hour periods and contain 40 features. This means that each of these features represents a frequency of visits to a location in a particular 3 hour window. The outliers were labelled as points in the data in which the person living with dementia was admitted to hospital or had an infection. The labelling is done by a clinical

---

[1] https://github.com/datamllab/tods/tree/benchmark
[2] https://www.spotseven.de/

monitoring team who have been part of the study and interact with the participants based on the observations and measurements collected by the in-home monitoring technologies integrated into a clinical dashboard. Only the people living with dementia who had at least one labelled outlier, were included in the final dataset. The data was labelled as an outlier $4$ days preceding and proceeding the infection and hospital admission episodes. Data points in which all of the feature values were $0$ and there was no hospital admission or infection were then removed. This leaves the contamination factor of the dataset at $2.3\%$. Graphs showing the distributions of values in the dataset can be viewed in Appendix A.2.2.

After applying the methods to the dataset, we split the data by each source (a household containing a person living with dementia) and tested the algorithms on each of these. We removed any sources in which there were less than $5$ outliers labelled. We will refer to this dataset as the separated Movement data.

The datasets chosen span a wide range of contamination values, allowing the algorithms to be evaluated over a range of scenarios. They also contain a mix of ratios of point outliers to collective outliers. See Appendix A.2 for more information.

We assumed that all of the datasets either have point outliers or collective outliers. Point outliers are defined by a single outlier on its own, whilst collective outliers are defined by consecutive outlier points in a time window.

## 2.2 MODELS

To test to what extent recurrent neural networks are more useful in detecting outliers within time series data, we have chosen three classical outlier detection methods as well as four recurrent neural networks set up in an outlier detection framework. All of these methods were chosen because of their ability to be useful for sequential outlier detection. The majority of the models were implemented using the TODS framework (Lai et al., 2021a), where the pipelines were built by Lai et al. (2021b). This was a requirement for us when deciding on which algorithms to evaluate as we required them to be readily available.

### EVALUATED METHODS

The methods include unsupervised techniques and recurrent models. The performance of the methods is based on reconstruction errors or the ability to separate unexpected patterns. These models use the error between the reconstructed or predicted values and the true values to identify the outliers.

A more descriptive list is given in Appendix A.3.

- **Stationary Methods**:
    - **OCSVM:** One-class support vector machines (OCSVMs) (Schölkopf et al., 1999).
    - **Isolation Forest:** (Liu et al., 2009).
    - **Auto-Encoder:** (Sakurada & Yairi, 2014) .
- **RNNs:**
    - **Autoregression:** (Rousseeuw & Leroy, 2005).
    - **GBRT:** Gradient boosting regression trees (GBRTs) (Friedman, 2001; Elsayed et al., 2021).
    - **LSTM-RNN:** Long short term memory recurrent neural networks (LSTM-RNNs) (Hochreiter & Schmidhuber, 1997; Bontemps et al., 2016; Singh, 2017).
    - **Self-Attention:** Descriptions of this paper's proposed methods are in the following section.

### PROPOSED METHODS

To get a better understanding of how recurrent neural networks perform on several time series datasets, we introduce two new methods based on the transformer model and self-attention mechanisms (Vaswani et al., 2017).

1. **Transformer Encoder:** This model uses positional encoding, self-attention and linear layers to calculate the next time point values based on a sequence of previous values. The loss between the predicted time point and the correct time point is used to calculate an outlier score using a Gaussian distribution.

2. **Transformer Encoder-Decoder:** This model uses a self-attention encoder to encode a sequence, before using a self-attention decoder to predict the next $x$ time points. The algorithm then calculates the mean loss over each of the predictions and uses a Gaussian distribution to calculate outlier scores. Within our experiments, we fix $x$ at 3.

Both of the above-mentioned models are a modification of the Long Short Term Memory Recurrent Neural Network (LSTM-RNN) outlier detection method. Instead of using an LSTM-RNN (Hochreiter & Schmidhuber, 1997; Bontemps et al., 2016; Singh, 2017) as the underlying time series forecasting model, we use transformer models (Vaswani et al., 2017) of two different architectures. A diagram of the proposed architecture is given in Appendix A.3. Unlike the LSTM-RNN outlier detection method, we do not focus on collective outliers, and so remove the sliding window which requires collective outliers to be prioritised (see Bontemps et al. (2016) for details).

One of the main draw-backs of LSTM-RNN is the way that it processes sequences. LSTM-RNNs process sequences one point at a time. This can cause problems when making predictions, since the model has to attempt to extract meaning from an input that was passed through the Long Short Term Memory (LSTM) block as many steps previously as the length of the sequence. Also, because of the way that they sequentially handle inputs, LSTM-RNNs can not be parallelised when calculating an output. Transformers aim to tackle this issue, along with introducing the self-attention unit (Vaswani et al., 2017). Transformers take entire sequences at once, allowing for all parts of the sequence to be understood by the model. This allows for further temporal relationships to be built. Moreover, because of this, transformers can be parallelised when training, and the encoding section can be parallelised when making predictions. We believe that a transformer based outlier detection technique would be a useful addition when attempting to find outliers in a dataset in which points can have complicated temporal dependencies within a sequence.

## 2.3 EVALUATION

To evaluate the effectiveness of our outlier detection methods on the datasets, we will mainly use the F1 score, which is defined as a harmonic mean of precision and recall. We will also look at the receiver operating characteristic (ROC) curve to understand the models performance against a random classifier.

We have tested each of the methods to understand whether Recurrent Neural Network (RNN) methods are required for outlier detection and if so, in which scenarios are they helpful. To answer this question, we have selected several datasets (both synthetic and real-world) that contain different types of outliers and span a wide range of contamination values.

We use contamination values of $5\%$, $10\%$, $15\%$, $20\%$ and $25\%$ throughout this paper. These are the values at which the outlier detection algorithm will be evaluated, however, they are not the values of the true numbers of outliers within the dataset.

In addition, the datasets vary in size and complexity (in terms of their underlying mechanics), allowing us to test the different individual algorithm's performance on collective outliers and point outliers that are global or contextual. We are especially interested in evaluating the methods on the Movement data, since this dataset contains complicated temporal relationships that may describe the outliers.

We will run each of the experiments 5 times, with various hyper-parameters (described in Appendix A.3), to make sure the results are reliable. Since in each experiment the model is detecting outliers on a single dataset, we train each of the forecasting models on the same data that we predict outliers on.

## 3 RESULTS

We start by looking at the results of the methods applied to the synthetic datasets supplied within the TODS package (Lai et al., 2021a). Figure 1a shows the maximum F1 score achieved on each of the synthetic datasets by either a non-recurrent or recurrent based outlier detection algorithm. These experiments show that non-recurrent based algorithms perform better (in terms of their F1 scores) than recurrent models when applied to the synthetic datasets, but only marginally. In fact, it could be argued that both types of methods produced similar results on these datasets.

The greatest F1 score was $90\%$ and was produced by the One Class Support Vector Machine (OCSVM) model. The median maximum F1 score (averaged over the datasets) achieved by recurrent methods was $61\%$ compared to the non-recurrent value of $63\%$. Figure 1b shows the number of times each algorithm was the best performance on a given dataset. This indicates that the non-recurrent based algorithms performed better than the recurrent models on datasets that were dominated by point outliers, however both performed similarly on collective outliers.

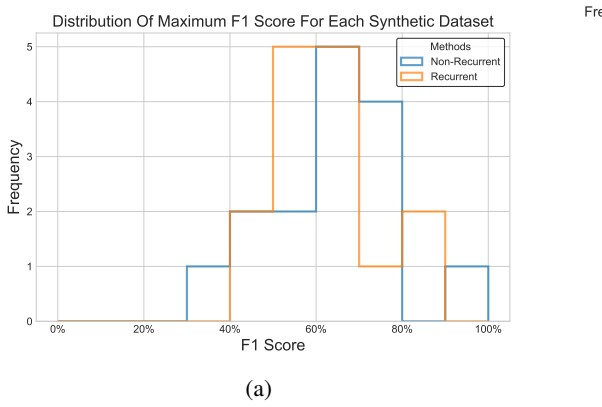
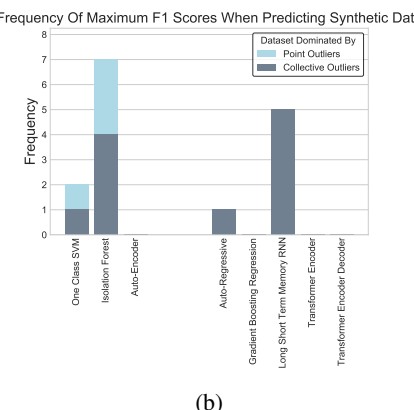

(a)

(b)

Figure 1: Figure 1a shows a histogram of the F1 scores that resulted from applying the methods to the different synthetic datasets 5 times (with the best performing contamination level for each model shown). Here, the maximum F1 score for each dataset is plotted for both the recurrent based methods and the non-recurrent based methods. Figure 1b shows the number of times a particular algorithm produced the best F1 score on a synthetic dataset. Auto-Endoer, GBRT, Transformer Encoder and Transformer Encoder-Decoder never showed the best performance on a dataset.

When applied to the real-world data supplied within the TODS package (Lai et al., 2021a), the non-recurrent outlier detection methods performed better. Figures 2a and 2b show the maximum F1 score for each of the methods applied to the real-world data. In both cases, the lowest performing non-recurrent method performed better than the highest performing recurrent method. When considering the SWAN-SF dataset in particular (Figure 2b), the non-recurrent methods had a greater advantage over the recurrent methods, with the best performing non-recurrent algorithm achieving an F1 score of $57.7\%$ compared to the best performing recurrent algorithm result of $39.2\%$.

However, the strong advantage the non-recurrent methods had on the real-world data is not present when considering the performance of the outlier detection algorithms on the Movement data. The F1 scores and receiver operating characteristic (ROC) curve can be observed in Figure 3. Here, the recurrent based algorithms and non-recurrent based algorithms performed similarly. The best result was obtained by a small margin, by the OCSVM model, which achieved a score of $5.77\%$ compared to the best performing recurrent algorithm (Transformer Encoder-Decoder) result of $5.58\%$. The ROC shown in Figure 3b, displays the true positive rate against the false positive rate and shows that both types of outlier detection algorithms performed better than random guessing (area of $> 0.5$ suggests better performance than random guessing), however neither performed confidently better than the other. Isolation Forest achieved the largest area under the receiver operating characteristic curve (AUC-ROC) with a value of $0.6$, which wasn't significantly larger than the greatest AUC-ROC value for the recurrent based algorithms (with a value of $0.59$, achieved by GBRT).

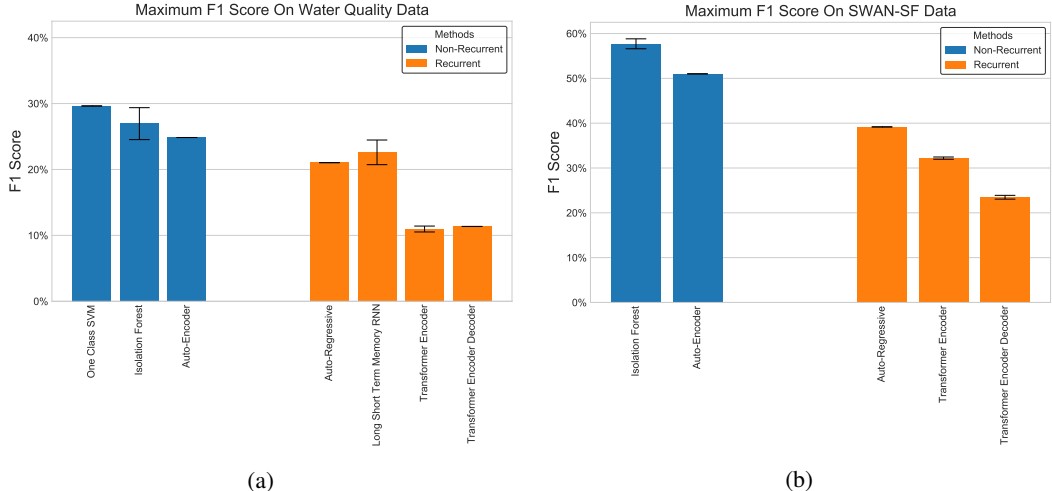

(a)                                                                        (b)

Figure 2: Figure 2a shows the average F1 score of the best performing contamination level for different methods when applied to the Water Quality dataset. Figure 2b shows the average F1 score of the best performing contamination level for the different methods when applied to the SWAN-SF dataset. Both of these datasets were discussed in Section 2.1.

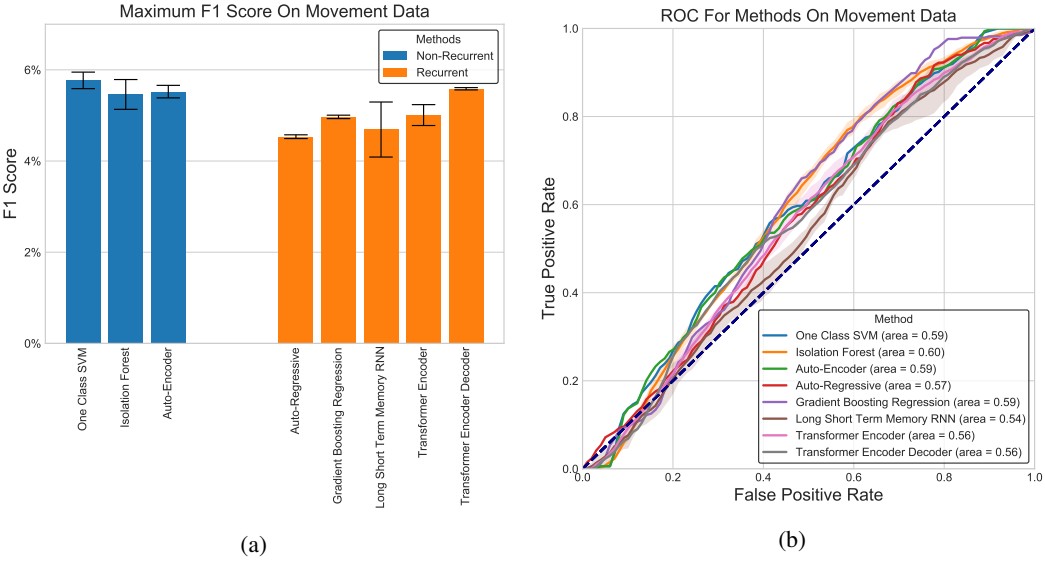

(a)                                                                        (b)

Figure 3: Figure 3a shows the average F1 scores of each model over 5 runs for the best performing contamination level on the Movement data. Figure 3b shows the ROC curve for these experiments in which the average F1 score was highest over the contamination levels.

The methods were then tested on the Movement data, split by source (each household containing a person living with dementia (PLWD)). Figure 4a shows the area under the ROC curve for each of the best performing recurrent and non-recurrent based algorithms when applied to the separated Movement data. These experiments show that the recurrent models performed significantly better than the non-recurrent methods (in terms of their AUC-ROC values) when applied to each of the households individually. The recurrent based algorithms achieved an average AUC-ROC of 73% compared to the average AUC-ROC of 60% achieved by the non-recurrent methods. We also see that all of the recurrent models (that achieved the maximum score on a given household's data) achieved an AUC-ROC value of larger than 50%, which is not the case with the non-recurrent based methods. This suggests that the recurrent models were on average always better than a random classifier, but the same is not true for the non-recurrent methods.

Table 1 shows the average maximum F1 score on each dataset by either a recurrent or non-recurrent based algorithm. We can see here, that the recurrent models were a significant improvement over the non-recurrent methods.

Table 1: Average maximum F1 score on each household's data, achieved by either a recurrent or non-recurrent method.

| Methods | Average F1 Score |
|---|---|
| Non-Recurrent | 12.9% |
| Recurrent | 19.1% |

Figure 4b shows the number of times each model scored the largest F1 score on a household with a PLWD. Here, it is clear that the recurrent based algorithms have an advantage, with the method producing the largest F1 score most often being the LSTM-RNN based outlier detection algorithm.

See Appendix A.4 for a small discussion on the effect of hyper-parameters.

These results show that recurrent based outlier detection algorithms make a significant improvement over the non-recurrent based algorithms when applied to the separated Movement data.

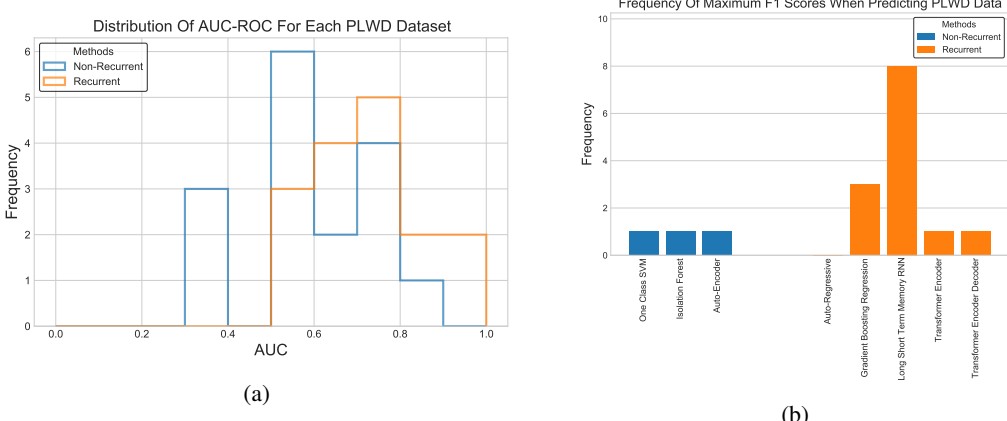

(a)

(b)

Figure 4: Figure 4a Shows a histogram of the values of the area under the ROC curve for each of the best performing recurrent and non-recurrent methods on the separated Movement data. Figure 4b shows the number of times each model obtained the best F1 score when applied to the separated Movement data. For both figures, each data point represents a result on a household's data.

## 4 DISCUSSION

From Section 3, we can see that when applied to the synthetic data, there were little differences between the performance of the two types of methods where the datasets were dominated by collective outliers. However, when point outliers were also considered, it was clear that the non-recurrent based outlier detection methods performed better. They also had the advantage of having fewer hyper-parameters to tune.

In addition, when applied to the two real-world datasets supplied in the TODS package (Lai et al., 2021a), non-recurrent based outlier detection methods perform better by achieving higher F1 scores. In the real-world tests on both datasets, the poorest performing non-recurrent methods still scored a larger maximum F1 score than the highest performing recurrent model.

On these experiments alone, it could be argued the benefit of applying recurrent models to outlier detection is not significant, since they are more complicated, have larger numbers of hyper-parameters, and perform less well than the non-recurrent methods. Although, it is worth noting that with more experiments and changes to the designs of the models; such as different hyper parameters, architecture changes, or different training pipelines, the recurrent based methods could offer improvements.

However, the results of the methods applied to the Movement Data were interesting, since both outlier detection methods struggled to produce meaningful results. No algorithm produced an F1 score larger than $6\%$ and the largest AUC-ROC was $60\%$. This is likely due to the way the dataset is constructed, since it is a concatenation of the movement data from several different households. Since each household's behaviour will be different, outliers caused by a given household will be disguised by another household's behaviour, unless the point is considered unusual behaviour when considering the entire cohort. This dataset is noisy and so the struggle in performance is understandable.

On the other hand, when the algorithms were applied to the separated Movement data (another real-world dataset), the recurrent methods performed significantly better. The main differences between the Water Quality and SWAN-SF datasets, and the Movement data is the complexity of the outliers. Outliers in the SWAN-SF and Water Quality datasets are categorised by sudden changes in the series [3] (Angryk et al., 2020b). Outliers in the Movement data are due to an individual's daily activities and behaviour and complex temporal relationships which could be subject to changes due to various factors such as seasonal changes, wellness and daily events. Figure 4 shows that the recurrent methods had a significant advantage in this scenario. This is likely because in this case, the recurrent based models were able to learn the individual behavioural signatures of the household they were searching for outliers within, enabling them to learn the complex temporal relationships that might make a point an outlier.

These experiments suggest that if outliers are hidden in complex temporal relationships, then recurrent based outlier detection algorithms can help classify them more accurately.

## 5 CONCLUSION

This paper focused on improving on the work done by Lai et al. (2021b) by asking a similar question to Elsayed et al. (2021). The goal was to understand whether recurrent based outlier detection algorithms were significantly better performing than non-recurrent outlier detection based algorithms and in which contexts they can add value. This paper has shown that recurrent based outlier detection algorithms are important when outliers depend on complex temporal relationships hidden within the data, as was the case with the separated Movement data. However, when searching for outliers that are not defined by these subtle relationships, non-recurrent methods can perform significantly better and are easier to implement.

The advantages of recurrent based methods should be carefully considered before being employed, since the scenarios in which they add value can be difficult to define; for example, how does a user know whether their data has temporal related outliers within their dataset? The models also come with more hyper-parameter tuning and less explain-ability in their results. It is worth first attempting to use non-recurrent based methods to see whether the results are satisfactory before introducing more complexity in the form of recurrent neural networks.

The recurrent methods for online outlier detection can be trained on a previous subset of the data, but allowed to continue to make predictions as data streams in. This means that the deep learning models could continually learn and to update their underlying forecasting models as the data or environment changes (e.g. to accommodate for seasonal and contextual changes).

It would also be useful to understand how to know whether a particular dataset will have complex, non-linear temporal relationships within the data, since it is not obvious when this is the case and how it might affect the detection of outliers. This highlights the importance of exploratory analysis in the datasets before designing the models or deciding on the hyper-parameter selection and training.

It would also be interesting to evaluate the transformer based outlier detection models with a sliding window, similar to the LSTM-RNN method as this might improve performance on collective outlier detection (where outliers are grouped together) and possibly the Movement data. However, this should be designed carefully to avoid overfitting.

In conclusion, the results here demonstrate when recurrent models could be effectively applied to time series outlier detection. We also show that on our Movement data, transformer based outlier detection models provide an effective way to detect complex temporal dependent outliers and make suggestions for further improvements.

---

[3] https://www.spotseven.de/wp-content/uploads/2018/03/rulesGeccoIc2018.pdf

## 6    REPRODUCIBILITY STATEMENT

These experiments are designed to be reproducible and easy to extend. Please see the IPython notebook titled "Testing Recurrent vs Non-Recurrent", available in the supplementary material. This notebook provides instructions on how to install the TODS package as well as the code required to run the experiments. The code itself saves all of the outlier scores for every test, allowing the user to quickly load and study them after running the algorithms.

In addition, this paper provides descriptions of the different algorithms in Appendix A.3 as well as clear details of hyper-parameters used. We also provide clear definitions of the metrics used and how they are calculated. We do not however, give access to the Movement data, since this is a sensitive dataset that forms part of a wider study within our centre. To this end, we do provide details on some of the statistics of the dataset (that can be viewed in Appendix A.2.2) as well as the preprocessing done (see Section 2.1).

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

# A APPENDIX

## A.1 METRICS

### F1 SCORE

Precision is defined as:

$$\text{Precision} = \frac{\text{True Positives}}{\text{True Positives} + \text{False Positives}}$$

And is interpreted as the proportion of predicted positives that were actually positive.

Recall is defined as:

$$\text{Recall} = \frac{\text{True Positives}}{\text{True Positives} + \text{False Negatives}}$$

This is the proportion of actual positives that were correctly classified as positive.

Finally, The F1 Score is defined as:

$$\text{Recall} = \frac{2}{\text{Recall}^{-1} + \text{Precision}^{-1}}$$

And is interpreted as the harmonic mean of the recall and precision. The larger the F1 score, the better a classifier is at classifying positives. The F1 score can take values between $0$ and $1$.

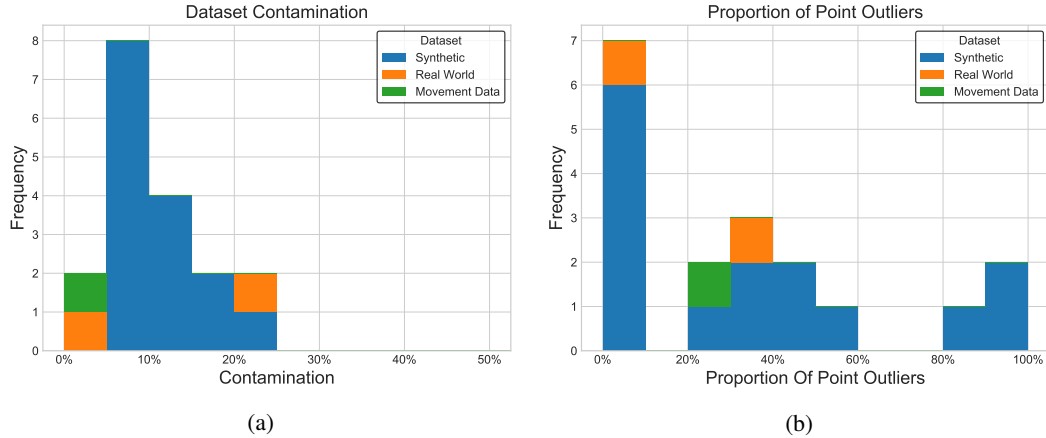

(a)                           (b)

Figure 5: Figure 5a shows the contamination of the synthetic data, the real-world data and the Movement data. Figure 5b shows the proportion of point outliers that are in each of these datasets. We assume all of the datasets either have point outliers or collective outliers.

COEFFICIENT OF VARIATION

The coefficient of variation is defined as:

$$c_v = \frac{\text{Standard Deviation}}{\text{Mean}}$$

This can take any positive value, with a larger value representing a higher variation in the series being tested.

## A.2 DATA

A histogram showing the distribution of contamination values for the different datasets is given in Figure 5a. Similarly, the proportion of point outliers to collective outliers is shown in Figure 5b.

### A.2.1 TODS PACKAGE REAL WORLD DATA

The TODS package contains four real world datasets, and scripts to load them. Unfortunately, when testing, we were only able to successfully perform testing on two of these. We believe, along with our own datasets, this is enough to understand the performance of the models.

### A.2.2 MOVEMENT DATA

Figure 6 shows the distribution of the values in the Movement data.

Figure 7 shows the average values in the Movement data, split by the time of day. Each data point that we calculate an outlier score for, is made by flattening an array like this into a single vector.

### A.3 OUTLIER DETECTION ALGORITHMS

Here, we give a brief description of each of the outlier detection algorithms that are tested within this paper:

- **Stationary Methods**:
    - **OCSVM:** One-Class Support Vector Machines (OCSVMs) (Schölkopf et al., 1999) are unsupervised outlier detection techniques which calculate a binary function that captures where the majority of the data lives, leaving a proportion, $P$ of the data outside of the positive region of the functions output. This proportion, $P$ corresponds to the contamination of the outliers in a dataset.

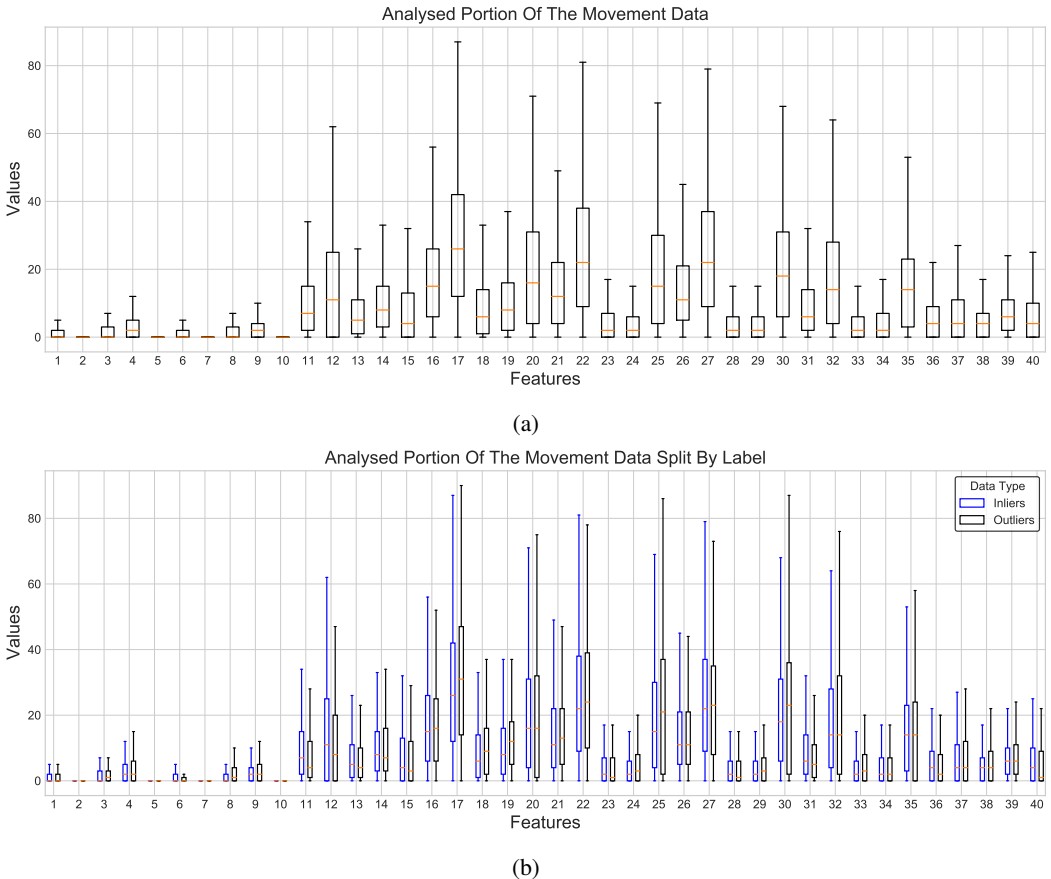

(a)

(b)

Figure 6: Box plot showing the distributions of values in the Movement data after processing. Figure 6a shows the distribution of the whole dataset, whereas Figure 6b shows the distribution split by their label.

Figure 7: This shows the average frequencies of visits to each location over the entire cohort, separated by time.

– **Isolation Forest:** Isolation Forest (Liu et al., 2009) is an unsupervised outlier detection technique that is built upon the concepts of random forest models. After building

a tree used to segment a dataset, a point is considered more of an outlier than another if it takes less branches in a tree to separate that point.

- **Auto-Encoder:** Using Auto-Encoders for calculating outlier scores (Sakurada & Yairi, 2014) relies on mapping points to a latent space that is used to reconstruct the data. Outliers are assumed to have more reconstructive loss than inliers, and this loss is used to score points as anomalies.

- **RNNs**
  - **Autoregression:** Autoregressive models assume that each point is linearly correlated to the previous few points (Rousseeuw & Leroy, 2005). In this way, this algorithm can be considered an RNN outlier detection method.
  - **GBRT:** Gradient Boosting Regression Trees (GBRTs) (Friedman, 2001; Elsayed et al., 2021) maps time series sequences into single time points by concatenating them along the feature axis. Regression is then used to predict the next time point and the loss is used to derive an outlier score.
  - **LSTM-RNN:** Long Short Term Memory Recurrent Neural Networks (LSTM-RNNs) take a similar approach to Autoregressive models. They assume that points can be predicted from the previous few points using an LSTM-RNN. The loss between the predicted time point and the actual time point are used to calculate the outlier score (Bontemps et al., 2016; Singh, 2017). In this calculation the LSTM-RNN outlier detection method employs a sliding window to check the density of outlier scores and detect collective outliers.
  - **Self-Attention:** Description was given in the Section 2.2. This model was built using the python package Pytorch (Paszke et al., 2019).

For each of the models, we tested various hyper-parameters and display the results of the best performing model over 5 repeated tests. These hyper-parameters mostly align with the parameters chosen in Lai et al. (2021b).

- **Stationary Methods**:
  - **OCSVM:** We tried both the subsequence (with window size of 10) and standard implementations of the model provided by the TODS framework.
  - **Isolation Forest:** We also tried both the subsequence (with window size of 10) and standard implementations of the model provided by the TODS package.
  - **Auto-Encoder:** We used feed forward neural networks with sequential layer sizes of (1,4,1), (1,4,8,4,1), (5,32,16,16,32,5), (32,16,16,32), (1,4,16,32,16,4,1). These were all tested with a dropout rate of 0.2, batch size of 32 and trained for 50 epochs. We also tested this method using the subsequence implementation in the TODS package.
- **RNNs**
  - **Autoregression:** We tested a step size of 1 and a window size of 5 and 10.
  - **GBRT:** We tested a step size of 1 and window sizes of 3, 5 and 10. We also tested this method using the subsequence implementation in the TODS package on each of the different window sizes.
  - **LSTM-RNN:** We tested 1, 2, 5 and 10 layers. For the single layer LSTM-RNN, we tested a hidden vector size of 1. For the layer sizes of 2, 5 and 10, we tested hidden vector sizes of 32 and 64. All other parameters were left as the default within the TODS package.
  - **Self-Attention:** For both the encoder and the encoder-decoder model, we tested models with 2 and 5 encoder and decoder layers as well as with feed-forward network dimensions of 32 and 64. These models were trained for 20 epochs, with a batch size of 30. The encoder only model used 5 data points to forecast the next data point, whereas the encoder-decoder model used 5 data points to forecast the next 3 data points.

A photo of the architecture implemented in the Self-Attention Encoder-Decoder model is given in Figure 8.

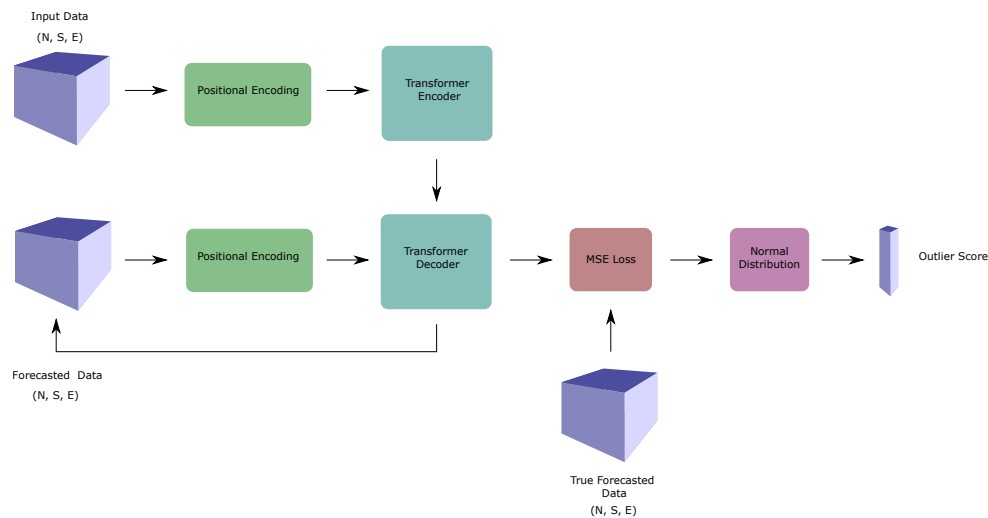

Figure 8: Self-Attention Encoder-Decoder outlier detection architecture.

## A.4 Effect of Hyper-Parameters

When testing the algorithms on the separated Movement data, we also calculated the median coefficient of variation (see Appendix A.1) for the F1 scores over each of the hyper parameter tests. This tells us how the F1 scores varied when different hyper-parameters were given. Table 2 shows the results. Interestingly, the largest variation in F1 score comes from the OCSVM model, and the lowest from the Transformer Encoder-Decoder, suggesting that good choices of hyper-parameters are needed for both recurrent and non-recurrent based outlier detection algorithms.

Table 2: Average coefficient of variation for the F1 scores produced by the different model hyper-parameters when applied to the separated Movement data. The smaller, the less the hyper parameter choices effected the results.

| Methods | Average $c_v$ |
|---|---|
| One Class SVM | 0.43 |
| Isolation Forest | 0.37 |
| Auto-Encoder | 0.33 |
| Auto-Regressive | 0.24 |
| Gradient Boosting Regression | 0.27 |
| Long Short Term Memory RNN | 0.31 |
| Transformer Encoder | 0.26 |
| Transformer Encoder Decoder | **0.23** |

Although these are interesting statistics, we were not able to test all reasonable hyper-parameters and there is no way of saying how comparable hyper-parameter changes were across models. For example, changing the OCSVM method from its default settings to a subsequence method is not necessarily comparable to changing the layer sizes in a Transformer Encoder-Decoder model.

