# OpenReview forum: "When Complexity Is Good: Do We Need Recurrent Deep Learning For Time Series Outlier Detection?"
_ICLR.cc/2022/Conference — ICLR 2022 Submitted_

### Official Review · Reviewer_cSV4 · 2021-10-29

**Correctness:** 2
**Technical Novelty And Significance:** 1
**Empirical Novelty And Significance:** 1
**Recommendation:** 3
**Confidence:** 4

**Main Review:**

Strengths:

1) Anomaly detection in time series is an important problem

Weaknesses:

1) The evaluation seems incomplete.

Details:

1) The title/abstract makes you believe that will read some other kind of paper but this is essentially an evaluation paper of anomaly detection methods. Nothing is wrong with that, this is just far from complete. There are dozens of methods to consider.

2) Not sufficient datasets to make such claims. Not sufficient analysis between anomaly types. Many questions of this form: are CNNs better? Are subsequence methods better? Are traditional vs modern approaches better? ....

**Summary Of The Paper:**

The paper evaluates a set of methods for time-series anomaly detection, including RNNs among the considered techniques. The evaluation on a set of datasets shows that RNNs perform better than the non-RNN architectures.

**Summary Of The Review:**

Preliminary results of an evaluation work for time-series anomaly detection. Motivation has to be reconsidered.

---

> ### Author Response · Authors · 2021-11-22
> **Reply to: Official Review of Paper3678 by Reviewer cSV4**
>
> Thank you for taking the time to review our paper and for your constructive comments.
>
> The aim of this paper was to understand where RNN based methods are useful compared to non-RNN based methods for time series outlier detection. We wanted to show the applicability of methods rather than perform model optimisation. Therefore, we chose the vanilla versions of models that we thought were representative of models used in practise. We also included subsequence models and tested various layer sizes and numbers, and many hidden vector sizes (for models that support such options).
>
> The use of more datasets to further this research is a good one and we will take this feedback on board.

---

### Official Review · Reviewer_FAw1 · 2021-11-02

**Correctness:** 2
**Technical Novelty And Significance:** 1
**Empirical Novelty And Significance:** 2
**Recommendation:** 3
**Confidence:** 5

**Main Review:**

The strengths of this paper are as follows:
- Time series outlier detection is an important problem as it can be applied to many real-world applications
- The research question 'do we need recurrent neural networks for time series outlier detection?' is of interest to the community
- A series of empirical results is presented to answer the question

The weaknesses include:
- The work is a rather simple comparative study, performing a few relevant models on some existing datasets without in-depth analysis of the experimental settings and empirical results. As a result, the empirical findings are not convincing. Some major issues here include: 1) there are a number of recent LSTM/RNN-based models that are specifically designed for time series outlier detection, but they are not used in the study; only basic LSTM/RNN models are used instead. 2) the way of constructing or injecting the outliers (as well as the types of outlier) can largely affect the performance of each model, but this is largely ignored in the study. 3) the analysis of the empirical results is so shallow that it cannot provide any insight into the underlying reasons of the obtained results.
- The crucial concept -- data complexity of time series data in outlier detection -- is not properly defined and quantified. Without a clear definition of data complexity, it is invalid to state whether or not more complex models are needed. The authors claim about the data complexity in terms of the presence of some complex outliers, such as collective outliers, but detailed analysis of this complexity through some ablation studies  is missing. There are many other complexity factors that should be considered, for example, how is the temporal dependence presented in each dataset? does the length of the temporal dependence vary significantly within each dataset? etc.
- It is unclear whether the used models are capable of detecting collective outliers or not. If the answer is yes to some models, how?
- No technical novelty is identified. It is a purely comparative study with only some shallow results. The work may be largely improved by including more datasets with natural real-life outliers and performing controlled experiments to justify some specific questions rather than some high-level general questions.

**Summary Of The Paper:**

This paper presents a comparative study of the performance of non-recurrent models and deep recurrent models for time series outlier detection. Seven models are evaluated on multiple synthetic and real-world time series datasets.

**Summary Of The Review:**

Overall, the work seems to be a technical report that contains some preliminary results but has many key issues to be further addressed.

---

> ### Author Response · Authors · 2021-11-22
> **Reply to: Official Review of Paper3678 by Reviewer FAw1**
>
> Thank you for taking the time to review our paper and for your constructive comments. We will take these comments on board for our future work.
>
> For our testing, we wanted to show how the applicability of each of the models to the different datasets, rather than focusing on model optimisation. We performed hyper parameter searches and tested several different architectures (varying numbers of layers and their sizes, and hidden vector sizes).
>
> Outliers were only injected into the synthetic datasets, which were left untouched from the TODS python package (https://github.com/datamllab/tods/tree/benchmark). Here, the outliers are injected in various ways and different mixes of these injection methods were tested.
>
> Understanding how to better quantify the complex temporal relationships that can exist in the data would be a good area of further research. We used the context of the data to make inferences about the complexity of these relationships.
>
> All the tested models were capable of detecting collective outliers, and some were specifically designed to. All the non-RNN presented were tested with and without a subsequence approach, with a window size of 10 tested.
>
> We also proposed an outlier detection algorithm based on an attention model which performed comparably to other outlier detection techniques. We could improve the results by optimising all the models further. However, this would have negated our aim in this paper. The main goal of this paper is to show while there is a lot of emphasis on hyper parameter optimisation and overall tuning of a model towards a problem, the choice of model should start from the problem and data not the other way around. To support our argument, we chose existing benchmark data and models (basing the choices of techniques and datasets on this benchmark) and then showed using our real-world data that the results of the benchmark paper (by Lai et al. (2021b), referenced) are not necessarily true for datasets containing datapoints with complex temporal relationships.

---

### Official Review · Reviewer_aJuJ · 2021-11-02

**Correctness:** 2
**Technical Novelty And Significance:** 2
**Empirical Novelty And Significance:** 2
**Recommendation:** 3
**Confidence:** 5

**Main Review:**

There are many existing non-RNN methods for time series outlier detection which can be found in e.g.

Manish Gupta, Jing Gao, Charu C. Aggarwal, Jiawei Han:
Outlier Detection for Temporal Data: A Survey. IEEE Trans. Knowl. Data Eng. 26(9): 2250-2267 (2014)

Those methods are for both point and collective outliers and should be included in the evaluation study.

The conclusion is drawn on 3 real-world data sets, one of which (Movement data) seems not publicly available. I find this not sufficient. For comprehensiveness, I would recommend to use existing relevant data sets from UCI Repository and optionally add artificial outliers.





**Summary Of The Paper:**

The paper compares non-RNN and RNN methods in the outlier detection context, and concludes that non-RNN approach is suited to point outliers while RNN approach is suited to collective outliers.

**Summary Of The Review:**

The technical contribution is incremental. Relevant existing non-RNN methods for outlier detection on time series are not included in the study.

---

> ### Author Response · Authors · 2021-11-22
> **Reply to: Official Review of Paper3678 by Reviewer aJuJ**
>
> Thank you for taking the time to review our paper and for your constructive comments.
>
> Thank you for the suggestion to include some more methods, discussed in
>
> Manish Gupta, Jing Gao, Charu C. Aggarwal, Jiawei Han: Outlier Detection for Temporal Data: A Survey. IEEE Trans. Knowl. Data Eng. 26(9): 2250-2267 (2014)
>
> In this work we wanted to show the applicability of different methods to different forms of data, rather than focusing on the model optimisation. That’s why we also chose the vanilla version of the studied models. However, the idea of including more datasets is a good suggestion. We will take this feedback on board for future work.
>
> We concluded that the RNN based methods make improvements when the dataset contains datapoints with complex temporal relationships. By doing this, we showed that the results of the benchmark paper (by Lai et al. (2021b), referenced) are not necessarily true for datasets containing datapoints with complex temporal relationships.

---

### Official Review · Reviewer_HFSk · 2021-11-07

**Correctness:** 3
**Technical Novelty And Significance:** 2
**Empirical Novelty And Significance:** 2
**Recommendation:** 3
**Confidence:** 4

**Main Review:**

The work presents numerous experimental results. However, the related literature should be further developed to get a better sense of the different methods existing. Additionally, some experimental clarification would help the reader understand: are the authors using a Monte Carlo cross-validation? Is the contamination repeated in each fold? How is the contamination performed?

Moreover, the results need statistical comparisons to understand if the distributions between recurrent and static models are significantly different (KS test) and also if the ROC are different (DeLong test). The current figures do not show clear performance gain from any method.

The final message should also be clarified: which method should be used in which context? Do some approaches perform better when lower dimensionality, high colinearities, long time patterns? Is one approach to be prefered if static methods are used. It would also be important to integrate an 'outlier time series' comparison to have a full overview of the use case of the different approaches.

Finally, the proposed approaches do not seem to offer any gain, it might be more interesting to develop on which methods perform better in which context.

**Summary Of The Paper:**

This paper studies time series outlier detection. The authors propose to compare recurrent and static outlier detection methodologies on multiple synthetic and real-world temporal datasets. Additionally, they present an extension to the LSTM outlier detection using an attention-based approach. The analysis suggests that non-recurrent methodologies should be prefered as they perform marginally better while requiring fewer parameters and training. Only in the context of complex temporal relations, recurrent approaches should be used.

**Summary Of The Review:**

The authors present a substantial experimental work that offers interesting insights on time series outlier detection. However, the paper needs further statistical analysis to better understand the significance of these results. Additional analyses would also help to understand which techniques to use in different contexts.

---

> ### Author Response · Authors · 2021-11-22
> **Reply to: Official Review of Paper3678 by Reviewer HFSk**
>
> Thank you very much for taking the time to review our paper and for your constructive comments. We will take these comments on board for our future work.
>
> We are not using Monte Carlo cross-validation. Since these methods are unsupervised, we don’t use any cross validation. Instead, we run each of the methods 5 times and calculate the statistics based on these runs. The same dataset and contamination are used for each of these 5 runs. Also, since our data was a time series, preforming a Monte Carlo cross-validation would not have respected the relationships between preceding and proceeding data points.
>
> The contamination process is different for each of the datasets. For the real-world datasets, the outliers are based on the labelling given in TODS (https://github.com/datamllab/tods/tree/benchmark) package and are context specific. For the Movement dataset, a point is considered an outlier if the corresponding person living with Dementia had a hospital admission or infection. For the synthetic data, the outliers were labelled based on the process outlined in the TODS package and the paper by Lai et al. (2021b), referenced.
>
> The contamination values are used by the models only, this number does not affect the outliers labelled in the dataset. By leaving the underlying data unchanged in this respect, we can more confidently compare the evaluation metrics across contamination values.
>
> The proposed method offered comparative results to the other methods and with more hyper parameter tuning, might perform better. However, this was not the purpose of the paper, since our focus was on when RNN based methods offer an advantage over non-RNN based methods in time series outlier detection. To investigate this, we used benchmark datasets and models (in their vanilla state, with some hyper parameter searching) that we thought were representative of those used in practise. By using our real-world data, we showed that the results of the benchmark paper (by Lai et al. (2021b), referenced) are not necessarily true for datasets containing datapoints with complex temporal relationships.

---

### Decision · Program_Chairs · 2022-01-20

**Decision:**

Reject

**Comment:**

This paper has been reviewed by four experts. Their independent evaluations were consistent, all recommended rejection. I agree with that assessment as this paper is not ready for publication at ICLR in its current form. The reviewers have provided the authors with ample constructive feedback and the authors have been encouraged to consider this feedback if they choose to continue the work on this topic.